# How Do Pharmacists Distribute Their Work Time during a Clinical Intervention Trial?—A Time and Motion Study

**DOI:** 10.3390/pharmacy12040106

**Published:** 2024-07-09

**Authors:** Renata Vesela Holis, Renate Elenjord, Elin Christina Lehnbom, Sigrid Andersen, Marie Fagerli, Tine Johnsgård, Birgitte Zahl-Holmstad, Kristian Svendsen, Marit Waaseth, Frode Skjold, Beate Hennie Garcia

**Affiliations:** 1Hospital Pharmacy of North Norway Trust, 9291 Tromso, Norway; 2Department of Pharmacy, UiT The Arctic University of Norway, 9037 Tromso, Norwaykristian.svendsen@uit.no (K.S.);

**Keywords:** emergency department, acute care, pharmacists, time distribution, observation, time and motion study, WOMBAT

## Abstract

Emergency departments (EDs) handle urgent medical needs for a diverse population. Medication errors and adverse drug events pose safety risks in the ED. Clinical pharmacists, experts in medication use, play a crucial role in identifying and optimizing medication therapy. The aim of this study was to investigate how clinical pharmacists introduced into the ED interdisciplinary teams distribute their work time. In a time and motion study, we used the Work Observation Method By Activity Timing (WOMBAT) to observe pharmacists in two Norwegian EDs. The pragmatic approach allowed pharmacists to adapt to ED personnel and patient needs. The pharmacists spent 41.8% of their work time on medication-related tasks, especially those linked to medication reconciliation, including documenting medication-related issues (16.2%), reading and retrieving written information (9.6%), and obtaining oral information about medication use from patients (9.5%). The remaining time was spent on non-medication-related tasks (41.8%), and on standby and movement (17.4%). In conclusion, ED pharmacists spent 42% of their work time on medication-related tasks, predominantly medication reconciliation. Their relatively new role in the interdisciplinary team may have limited their broader clinical impact. Relative to other ED healthcare professionals, ED pharmacists’ goal remains to ensure accurate patient medication lists and appropriate medication use.

## 1. Introduction

The emergency department (ED) is a complex and dynamic setting that provides urgent and life-saving care to patients with diverse and unpredictable healthcare needs. EDs are staffed by interdisciplinary teams of healthcare professionals, such as physicians, nurses, paramedics, and technicians, who aim to deliver high-quality and safe care [1]. The team configuration varies across countries, particularly regarding the integration of pharmacists. Few countries offer specific training for ED pharmacists [2].

Medication-related tasks are essential for ensuring patient safety and quality of care throughout the healthcare system [3]. These tasks include medication reconciliation (MedRec), medication review (MedRev), prescribing, dispensing, administration, monitoring and education. Paradoxically, these tasks are often neglected or delayed in EDs, where patients are at considerable risk of medication errors and adverse drug events [4]. The standard ED healthcare personnel, physicians and nurses, are frequently overwhelmed by their high workload and critical patient care and so tend to prioritize other clinical tasks over purely medication-related tasks [5]. Both physicians and nurses spend the majority of their time on non-medication-related tasks [6,7,8]. Physicians, who are the main personnel responsible for patient safety, experience that MedRec and MedRev are time consuming and that medication-related tasks represent a cognitive burden [5]. As a result, many medication-related issues remain unresolved or unaddressed in the ED, which may postpone correct diagnostics, prolong the time in hospital, impair patient recovery and increase the risk of harm during the hospitalization or after discharge [9,10]. Despite this knowledge, few EDs are equipped with ED pharmacists, ensuring high-quality medication safety in acute care [11].

Clinical pharmacists are experts in medicines and their use, and they may through various medication-safety services, such as MedRec, MedRev, medication counseling, patient education and antimicrobial stewardship, optimize medication therapy and resolve medication-related problems [12,13]. Several studies from different countries have shown that ED pharmacists improve medication safety, reduce hospitalization, enhance patient satisfaction and save costs [14,15,16,17]. The pharmacists’ contribution to the interdisciplinary team can also relieve the other team members of their medication-related tasks and other duties, and reduce their work and cognitive overload [18,19]. 

In Norway, as in many countries worldwide, the ED pharmacists’ role is not well established. Few EDs have access to an ED pharmacist, and no specialist education or training exist within ED pharmacy care. Furthermore, there is a lack of research on how clinical pharmacists in Norway impact patient care and patient-related outcomes [20]. The multicenter trial Pharmacist in ED (PharmED) was conducted to investigate the effects of introducing clinical pharmacists into the interdisciplinary ED team in three EDs in North Norway [21]. 

The aim of this paper was to investigate how the pharmacists distributed their work tasks during the intervention period in two of the EDs.

## 2. Materials and Methods

### 2.1. Study Design and Setting

In this time and motion study, we used the Work Observation Method By Activity Timing (WOMBAT) during the intervention period [22]. We observed pharmacists in the two largest study EDs (ED1 and ED2) during November 2021–January 2022, when the pharmacists had been working there for seven and three months, respectively. The third, and smallest, ED in the PharmED trial was excluded because it had only two pharmacists sharing the position and therefore could not be sufficiently anonymized.

ED1 and ED2 annually admit about 15,000 and 12,000 patients, respectively. They usually operate without pharmacists, as their standard teams consist of physicians, nurses and healthcare secretaries. The patients are only admitted to the EDs if they have referral from their general practitioner or from a municipal urgent care clinic providing ambulatory care outside of general practitioner office hours. The ED healthcare personnel provide immediate help, conduct physical examinations and obtain diagnostic tests. It is the physicians’ responsibility to prescribe and discontinue medications, establish a medication list and to perform MedRec and MedRev. Furthermore, the physicians determine whether the patient will be admitted to a hospital ward or discharged.

### 2.2. The Pharmacist Intervention

The eleven pharmacists integrated in the two ED teams were trained clinical pharmacists with experience in other departments. Only two had ED experience prior to the PharmED study, one in Norway and one in Scotland. Prior to the PharmED study, the pharmacists had undergone a short training program, with lectures, seminars, discussions and observations, to become acquainted with the EDs’ personnel and work flow. The majority of the included pharmacists, as well as the ED physicians and nurses, had not previously engaged in any collaborative efforts in an ED setting. Among the eleven pharmacists integrated into the two EDs, we observed ten.

The introduction of ED pharmacists followed a stepped-wedge design, commonly used for complex interventions in healthcare settings. This design allows the intervention to be implemented sequentially, enabling control for differences between study sites and for long-term effects throughout the study period, while also minimizing spillover effects. In our study, the rollout occurred at three-month intervals: starting on 3 May 2021 in ED1, followed by 2 August 2021 in ED2, and 1 November 2021 in ED3. The intervention lasted until 31 January 2022. Consequently, the pharmacists worked for nine, six and three months in ED1, ED2 and ED3, respectively [21]. ED1 and ED2 were equipped with two daily pharmacist shifts, Mondays through Fridays. The first from 08:00 a.m. to 3:30 p.m., and the second from 11:30 a.m. to 07:00 p.m., ensuring overlap during the busiest hours. Five and six pharmacists shared the shift work in ED1 and ED2, respectively. 

All the pharmacists were well-acquainted with working in the respective hospitals, and they were asked to deliver their services according to the needs and preferences of ED healthcare professionals and patients. The pharmacists were encouraged to explore and adapt to the unfamiliar environment, and to find their own ways of contributing to the team. The pharmacists had to identify and seize opportunities where they could add value and improve their professional services. Likewise, the ED physicians and nurses were instructed to utilize the pharmacists’ expertise and skills.

### 2.3. Data Collection Tool and Piloting

The validated WOMBAT software was used to collect data [22]. This software is specifically designed for direct observation of healthcare professionals’ work and communication patterns, and it enables structured recording of the multi-dimensional features of these patterns. The WOMBAT software runs on a tablet and automatically time stamps the tasks entered, thus capturing the duration of task performance, as well as interruptions and multi-tasking [23].

The WOMBAT tool consists of four dimensions: *what, where, with whom,* and *how*. The *“what”* dimension describes the type of activity, such as oral communication or documentation. The *“where”* dimension indicates the location of the activity, such as in the ED, outside the ED, or in the medicine room. The *“with whom*” dimension specifies the interaction partner of the observed person, such as a patient, physician, nurse, or ED pharmacist. The “*how*” dimension defines the mode of communication, such as face-to-face or by phone, or the tool used for the activity, such as patient electronic medical record, medication list, or online medication interactions search tool. The observer must record at least one *“what”* and one *“where”* category for each activity but can leave the “*with whom*” and “*how*” dimensions blank when not applicable. The observer can also record any other relevant activity as multitasking or interruptions, depending on what is observed. The activities are further classified into the medication-related or non-medication-related sub-categories, except for the categories *movement* and *standby*. If the pharmacists conducted work related to other hospital departments while working in the EDs, those activities were recorded as *other* in the *“what”* dimension.

Three researchers (M.F., T.J. and R.V.H.) had previously developed observation categories and adapted the WOMBAT tool to the study settings by shadowing and interviewing physicians and nurses in the EDs [7,8]. They identified and defined the work tasks, locations, interactions and tools involved, which finally resulted in 53 mutually exclusive categories of interest, for which inter-rater validation was performed [7,8]. The current study’s observer (S.A.) interviewed and shadowed the ED pharmacists after they had been working in the EDs for some months and adjusted the WOMBAT categories for ED pharmacists. The adjustments were minor and included, e.g., removing a category “patient examination/treatment”. An experienced WOMBAT user and researcher (E.C.L.) reviewed and pilot tested the initial categories, after which final adjustments were made to the categories and definitions. The testing continued until all the pharmacists’ work tasks were clearly defined and could be observed without ambiguity. The final categories of the “*what*” dimension are shown in Figure 1, while all of the categories are available in Appendix A.

### 2.4. Data Collection

Observations were carried out from 30 November 2021 to 10 January 2022 in ED1, and from 8 November 2021 to 29 November 2021 in ED2. At this point, the ED pharmacists had been working for seven and three months in their respective EDs. Based on the pharmacist’ working schedule, the observations were carried out from Monday to Friday between 08:00 a.m. and 07:00 p.m.

Observations were carried out based on a predefined observation schedule designed to ensure the equal distribution of observations between days of the week and between times of the day. The observation sessions lasted from one to three hours, while most were two hours long. The observation time per day did not exceed six hours (three two-hour sessions) to avoid exhaustion of the observer [24,25].

### 2.5. Data Analysis

Data were collected using an iPad Mini^®^ with the WOMBAT app version 3.0 installed. Data were transferred from WOMBAT to Microsoft Excel^©^ version 2014 and then further managed and analyzed by SAS^®^ 9.4 software. The proportion of time spent on each task was calculated and presented as the % of the total observed time on each task category and sub-category. The sum of the proportions exceeds 100% due to multitasking. The 95% confidence intervals (CIs) were calculated by a bootstrap approach using SAS Macro programs developed for WOMBAT data [26]. Statistically significant differences were defined as non-overlapping 95% CIs.

### 2.6. Ethics

This study was approved by the Patient Protection Officer at the Hospital Pharmacy of North Norway Trust (No. 02330) and the involved hospitals.

## 3. Results

A total of 100 observation sessions were conducted in the two EDs, amounting to 196 h 55 min and 33 s (see Table 1 for details). During the observation time, the pharmacists assessed a mean of 5.8 patients per session and spent a mean of 14 min and 39 s per patient per session (see Table 1).

### 3.1. Distributions of Pharmacists’ Work Time per Task

The work tasks were categorized into 19 *“what”* categories and sub-categories, and 7 of them were identified as medication-related, as shown in Table 2. The medication-related work tasks accounted for 41.8% of the total observed time, the non-medication-related work tasks for 41.8%, while standby and movement accounted for 17.4% (see Table 2).

Among the medication-related tasks, the most time-consuming tasks included documentation (16.2%), reading and retrieving written information (9.6%), oral communication and retrieving information about medication use from patients, relatives, institutions, or pharmacies (9.5%), and communication about medications with healthcare personnel or patients (6.8%). No significant differences between the two EDs were observed.

Regarding the non-medication-related tasks, the pharmacists mainly engaged in reading and retrieving written information (17.9%), oral communication (8.3%), and logistics (4.5%). We identified a significant difference between the EDs in the time spent on non-medication-related logistics (5.2%, 95% CI 4.7–5.9 in ED1 vs. 3.8%, 95% CI 3.3–4.2 in ED2) and documentation (2.6%, 95% CI 1.7–3.6 in ED1 vs. 0.8%, 95% CI 0.5–1.2 in ED2) (see Table 2). We also identified a significant difference between the two EDs in terms of the standby time, for which the ED1 pharmacists spent 6.9% (95% CI 4.9–9.4) and the ED2 pharmacists 20.5% (95% CI 17.4–23.9) of their work time.

### 3.2. Where the Pharmacists Were

The pharmacists in both EDs spent most of their time in the EDs (Table 3). However, we identified a significant difference between the two EDs, where the ED2 pharmacists spent 97.2% (95% CI 92.9–102.5) of their time in the EDs, while the ED1 pharmacists spent only 79.2% (95% CI 75.8–82.7) of their time in the ED (see Table 3). This was mostly due to the ED1 pharmacists spending 20.1% of their time outside the ED.

### 3.3. Face-to-Face Interactions

Face-to-face interactions included all personal encounters, such as oral medication and non-medication-related communication or meetings, and represented 22% of the pharmacists’ time in the two EDs (see Table 4). Most of this time was spent with patients and relatives (7.8%), physicians (7.4%) or other pharmacists (6.5%). Compared with ED2, the ED1 pharmacists spent significantly more time interacting on a face-to-face basis with fellow pharmacists (9.1%, 95% CI 8.3–10.0 vs. 3.6%, 95% CI 3.1–4.2) and nurses (4.4%, 95% CI 3.8–5.0 vs. 2.8%, 95% CI 2.3–3.3). Conversely, the observed time spent on face-to-face interactions with physicians was significantly lower in ED1 (6.5%, 95% CI 5.7–7.3) compared to ED2 (8.4%, 95% CI 7.6–9.3).

## 4. Discussion

This is the first time and motion study providing a comprehensive analysis of the tasks undertaken by clinical pharmacists in an ED setting, along with the quantification of the time distributed to each work task. The pharmacists spent 41.8% of their time on medication-related tasks, 41.8% on non-medication-related tasks, and the remaining 17.4% on standby and movement. Additionally, we identified significant differences between the two involved EDs related to the time outside the EDs and face-to-face communication with the other healthcare personnel. 

Our results show that pharmacists spent 42% of their work time to MedRec, MedRev, and other medication-related activities, which are crucial for optimizing patient outcomes and facilitating care transitions [27,28,29,30]. This is far more than the time that both ED nurses and physicians, who devoted only 3.3% and 8.7%, respectively, to such tasks [7,8]. Consequently, pharmacists are the healthcare professionals in the ED dedicating most of their time to medication-related tasks. This aligns with expectations, considering their expertise in this field. Additionally, pharmacists also spent 42% of their work time on non-medication-related tasks, e.g., becoming acquainted with patient medical histories, medical records or laboratory data, which represents essential knowledge for conducting proper evaluations of patient medications. This implies that over 80% of the pharmacists’ time was committed to activities that directly or indirectly influence the pharmacists’ ability to work with optimizing medication use in the EDs. Introducing pharmacists into the ED setting emerges as a strategic move to improve medication-related patient care [31]. By dedicating the majority of their work time to these tasks, as also evidenced by this study, pharmacists have the potential to significantly improve the quality and safety of patient care, strengthen interdisciplinary collaborations, and positively impact patient satisfaction and the well-being of ED healthcare personnel [18,19,32].

We observed that MedRec was the predominant task on which the study pharmacists spent the largest proportion of their time. Despite that, they were granted autonomy and encouraged to expand their scope beyond MedRec responsibilities, e.g., MedRev, patient counselling and educational [2,21,33], and this finding was not unexpected. The pharmacists’ expertise in MedRec, supported by extensive experience, may have made their transition to a new department smoother. Their role in MedRec, expected to ease the medication-related workload of physicians and nurses, was a logical step forward. This is further underscored by the fact that MedRec is often perceived by ED physicians as both time-consuming and a detective job [5], which may have motivated physicians to readily delegate this responsibility to the pharmacists. We did, however, anticipate greater pharmacist engagement in broader clinical discussions and patient counselling, yet we observed a minimum of such interactions. This may also be attributed to the recent initiation of their roles and the lack of specialized ED training that is standard in countries like the USA and UK [34]. Addressing these educational gaps and fostering more diverse clinical involvement are crucial steps for the continued integration of pharmacists into ED settings.

Prior research indicates that physicians may not be fully aware of pharmacists’ expertise [5,35], which can complicate pharmacists’ integration into the ED interdisciplinary team. Effective integration requires mutual respect and comprehension of each other’s roles and competencies within the team. A crucial element for this is good communication across professions. We found that the majority of the pharmacists’ work time was solitary, with only 22% involving face-to-face interactions. Of these interactions, one-third were with patients. We identified a significant difference in pharmacist-to-physician face-to-face interactions between ED1 (29%) and ED2 (39%). Furthermore, there was a significant difference in pharmacist-to-pharmacist face-to-face interactions. In ED1, these accounted for the majority (40%) of all face-to-face interactions, while in ED2, they were significantly lower (17%). Interactions with nurses were considerably less frequent in both EDs. The differences between the EDs can be caused by various factors, such as the pharmacists’ placement within the EDs, the ease of access for physicians and nurses, or opportunities for pharmacists to consult with a fellow pharmacist on complex cases. However, it may also be related to how much time the pharmacists spent in the ED compared to outside. In ED1, the pharmacists split their time between the ED and other wards, whereas in ED2, they concentrated their efforts within the ED. This may have influenced both the face-to-face interactions and pharmacist–physician collaboration. The nature of ED work, characterized by both quiet and hectic periods, can facilitate the social integration of new team members. Engaging socially during quieter times can positively impact team dynamics [36]. An increased presence in the ED is likely to enhance communication and professional relationships, contributing to more integrated role within the ED team [37].

### Strengths and Limitations

This study has several strengths, including applying the structured and validated WOMBAT methodology and ensuring exact collection of observed task time applying several observation categories simultaneously [24]. Also, the meticulously developed observation categories, the inter-rater validation, and the number and duration of the observations strengthen the internal validity of our findings [38,39,40]. The limitations include observation bias due to the Hawthorne effect, i.e., people changing their behavior when they know they are being watched. This could have influenced how the pharmacists and other ED staff worked during the observation periods [41]. However, this should not be a major concern in our study, as previous studies have demonstrated that the Hawthorne effect is minimal when using the WOMBAT in critical care environments [22]. Cautiousness is, however, necessary when interpreting the significant results. A large number of statistical tests were performed, potentially causing a multiple testing problem. Another limitation is the generalizability of our findings to other settings and countries. The EDs in our study may have different characteristics and challenges to those in other regions and countries, and the role and scope of clinical pharmacy practice may vary depending on regulations and policies. Consequently, our results may not be directly applicable to other contexts without further validation and adaptation. Finally, our results are limited by the intervention study setting and the possibility that the pharmacists may not have been fully integrated in the team when the study was conducted. Future studies should explore the time distribution in a fully operational team.

## 5. Conclusions

This study shows that ED pharmacists in the PharmED study spent 42% of their work time on medication-related tasks. Predominantly, their medication-related activities were focused on medication reconciliation. Despite their preparedness and capability, the broader clinical influence of pharmacists on other medication-related tasks may be limited by educational gaps and the infancy of their roles. Addressing these challenges and promoting diverse clinical involvement are key to enhancing patient care and interdisciplinary collaborations. Nonetheless, pharmacists stand out as the healthcare professionals in the ED who devote most of their time to ensuring medication safety. The strategic integration of pharmacists, considering local ED contexts, is essential for optimizing their contribution to healthcare outcomes.

## Figures and Tables

**Figure 1 pharmacy-12-00106-f001:**
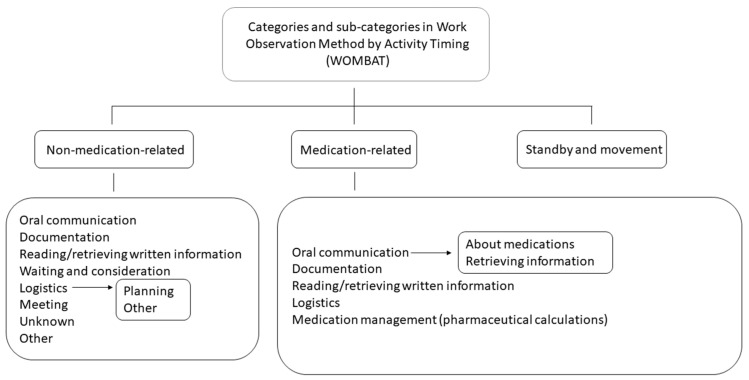
Overview of the “WHAT” categories and sub-categories in the WOMBAT application.

**Table 1 pharmacy-12-00106-t001:** Observation time, number of pharmacists observed and number of patients in each observation session.

	EDs Total	ED1	ED2
Total observed time, hh:mm:ss	196:55:33	100:38:12	96:17:21
Number of pharmacists observed	10	4	6
Number of sessions observed	100	51	49
Mean number of patients per session (min–max)	5.83 (0–13)	5.90 (0–13)	5.76 (1–11)
Mean time per patient per session in hh:mm:ss (min–max)	00:14:39 (00:00:37–01:54:54)	00:13:58 (00:01:51–00:51:20)	00:15:20 (00:00:37–01:54:54)

ED—emergency department; min—minimum; max—maximum.

**Table 2 pharmacy-12-00106-t002:** Pharmacists’ work time distribution overall and by emergency department (ED).

		EDs Total		ED1		ED2	
Total observed time, hh:mm:ss		196:55:33		100:38:12		96:17:21	
		**%**	(95% CI)	%	(95% CI)	%	(95% CI)
Medication-related		41.8		42.8		40.8	
Documentation		16.2	(14.6–17.7)	15.2	(13.3–17.5)	17.2	(15.1–19.6)
Oral communication		16.3	(14.9–17.8)	16.7	(14.3–18.7)	15.9	(13.9–18.3)
	Retrieve information about medication use	9.5	(8.2–10.9)	10.7	(9.1–12.6)	8.3	(6.5–10.5)
	About medications	6.8	(6.0–7.6)	6.0	(4.9–7.1)	7.7	(6.7–8.8)
Reading/retrieving written information		9.6	(8.8–10.4)	9.1	(8.2–10.3)	10.0	(8.9–11.4)
Logistics		0.1	(0.0–0.3)	0.2	(0.0–0.5)	0.0	-
Pharmaceutical calculations		0.1	(0.0–0.1)	0.1	(0.0–0.1)	0.1	(0.0–0.1)
Non-medication-related		41.8		47.0		36.3	
Reading/retrieving written information		17.9	(16.6–19.1)	19.7	(17.9–21.5)	16.0	(14.5–17.9)
Oral communication		8.3	(7.6–9.1)	9.3	(8.1–10.7)	7.3	(6.6–8.1)
Logistics		4.5	(4.1–4.9)	**5.2**	**(4.7–5.9)**	**3.8**	**(3.3–4.2)**
	Planning	4.4	(4.1–4.8)	**5.1**	**(4.6–5.6)**	**3.8**	**(3.3–4.2)**
	Other logistics	0.1	(0.1–0.2)	0.1	(0.1–0.2)	0.1	(0.1–0.2)
Meeting		3.3	(2.0–5.1)	4.2	(1.9–7.3)	2.4	(1.2–4.3)
Other		3.1	(1.9–4.9)	3.6	(1.4–7.0)	2.6	(1.5–4.2)
Waiting/consideration		2.2	(1.9–2.5)	1.8	(1.5–2.3)	2.5	(2.1–3.0)
Documentation		1.7	(1.3–2.2)	**2.6**	**(1.7–3.6)**	**0.8**	**(0.5–1.2)**
Confidential		1.2	(0.5–2.2)	1.2	(0.2–2.7)	1.2	(0.3–2.6)
Standby and movement		17.4		11.0		24.1	
	Standby	13.6	(11.5–15.5)	**6.9**	**(4.9–9.4)**	**20.5**	**(17.4–23.9)**
	Movement	3.8	(3.5–4.1)	4.1	(3.7–4.6)	3.5	(3.2–4.0)

CI, confidence interval; ED, emergency department; figures in bold indicate that the confidence intervals do not overlap, i.e., significant differences. The sum of proportions exceeds 100% due to multitasking.

**Table 3 pharmacy-12-00106-t003:** Distribution of time per location where the pharmacists spent their time by emergency department (ED).

	EDs Total		ED1		ED2	
Total observed time, hh:mm:ss	196:55:33		100:38:12		96:17:21	
	%	(95% CI)	%	(95% CI)	%	(95% CI)
In the ED	88.0	(85.2–91.2)	**79.2**	**(75.8–82.7)**	**97.2**	**(92.9–102.5)**
Outside the ED	11.4	(9.2–14.1)	**20.1**	**(15.4–24.8)**	**2.3**	**(1.5–3.4)**
In the COVID-19 room	0.4	(0.2–1.0)	0.5	(0.3–0.9)	0.4	(0.0–0.8)
In the medicine room	0.1	(0.0–0.3)	0.2	(0.0–0.6)	0.0	(0.0–0.1)

CI, confidence interval; ED, emergency department. Figures in bold indicate that the confidence intervals do not overlap.

**Table 4 pharmacy-12-00106-t004:** Mean proportion of time (%) spent by pharmacists on face-to-face interactions with patients or team members, total for both emergency departments (EDs) and for each ED.

		EDs Total		ED1		ED2	
Total observed time, hh:mm:ss		196:55:33		100:38:12		96:17:21	
		%	(95% CI)	%	(95% CI)	%	(95% CI)
In total		22.0	(21.2–22.9)	22.7	(21.4–23.9)	21.4	(20.2–22.6)
	With patients/relatives	7.8	(7.2–8.4)	7.8	(7.1–8.7)	7.7	(7.0–8.6)
	With physicians	7.4	(6.9–8.0)	**6.5**	**(5.7–7.3)**	**8.4**	**(7.6–9.3)**
	With pharmacists	6.5	(6.0–7.0)	**9.1**	**(8.3–10.0)**	**3.6**	**(3.1–4.2)**
	With nurses	3.6	(3.2–4.0)	**4.4**	**(3.8–5.0)**	**2.8**	**(2.3–3.3)**
	With others	2.6	(2.3–2.9)	2.9	(2.4–3.4)	2.3	(1.8–2.7)

CI, confidence interval; ED, emergency department. Figures in bold indicate that the confidence intervals do not overlap. The sum of proportions exceeds the total value due to multitasking.

## Data Availability

The data supporting this study’s findings are available upon reasonable request from the corresponding author, R.V.H.

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
