# Peer review of "How Do Pharmacists Distribute Their Work Time during a Clinical Intervention Trial?—A Time and Motion Study"

_pharmacy, 2024, doi:10.3390/pharmacy12040106_

Round 1

Reviewer 1 Report

Comments and Suggestions for Authors

Thank you for the opportunity to review this manuscript. I was very excited to read a time in motion study, as this methodology has always fascinated me. I have a number of comments to help ensure readers fully appreciate what the authors have done. I have arranged the comments by section. 

Introduction - no comments

Materials and methods:

Did the authors seek and receive IRB/ethics approval for this project? It would be helpful to have that added to the manuscript (I didn't seen this evidence). 

Pg 3, line 95 - The authors state that this was a stepped-wedge design, but provided very little detail for how this was done. While I appreciate it the protocol was published elsewhere, some details are needed in this manuscript for the reader to evaluate the findings accurately. Please transfer some of the specifics around why this design was chosen and how it was implemented in each of the EDs.

Pg 4, line 146-147 - Can the authors comment on why the observation periods for the EDs were so different from each other?

Results - no comments

Discussion:

Pg 8, line 224-225 - I am not sure I understand how the data presented demonstrates that pharmacists were given the autonomy to expand the scope of their work?

Pg 8, line 226-232 - Suggest removing these lines as they are not currently supported by the data collected and don't add substantively to the stated aim of the project. 

Pg 8, line 238-253 - Suggest moving this paragraph up to be second in the discussion section. 

Pg 9, line 266-270 - Suggest removing these two sentences as they do not add to the stated aim of the paper. 

Conclusion - no comments

Author Response

Dear reviewer, I sincerely appreciate your thorough review of our manuscript. Your valuable comments and suggestions helped me to improve the quality of our article. Thank you for your time and effort.

Did the authors seek and receive IRB/ethics approval for this project? It would be helpful to have that added to the manuscript (I didn't seen this evidence). 

Thank you for your feedback. Indeed, we have sought and received an ethical approval, which is detailed in the Institutional Review Board Statement at the end of the manuscript. Nonetheless, we agree that this information should be mentioned earlier in the body of the manuscript. Consequently, we have now included an Ethics subsection in the Methods section as point 2.6. “2.6. Ethics The study was approved by the Patient Protection Officer at the Hospital Pharmacy of North Norway Trust (No. 02330) and the involved hospitals.” 

Additional information regarding written consent is stated in the Informed Consent Statement section. 

Pg 3, line 95 - The authors state that this was a stepped-wedge design, but provided very little detail for how this was done. While I appreciate it the protocol was published elsewhere, some details are needed in this manuscript for the reader to evaluate the findings accurately. Please transfer some of the specifics around why this design was chosen and how it was implemented in each of the EDs.

Thank you for your suggestion. The clarification you recommended has now been incorporated into the manuscript (section 2.2. The pharmacist intervention): “The introduction of ED pharmacists followed a stepped-wedge design, commonly used for complex interventions in healthcare settings. This design allows the intervention to be implemented sequentially, enabling control for differences between study sites and for long-term effects throughout the study period, while also minimizing spillover effects. In our study, the rollout occurred at three-month intervals: starting on May 3rd, 2021, in ED1, followed by August 2nd, 2021, in ED2, and November 1st, 2021, in ED3. The intervention lasted until January 31st, 2022. Consequently, pharmacists worked for nine, six and three months in ED1, ED2 and ED3, respectively.”

Pg 4, line 146-147 - Can the authors comment on why the observation periods for the EDs were so different from each other?

Thank you for your inquiry.
The variation in the length of observation was primarily due to the scheduling and location constraints of our observer, SA, as well as the strategic timing of data collection to ensure quality and consistency.

SA was based long-term in the city of ED1, which allowed for a more extended and flexible observation schedule. In contrast, the observation period for ED2 was shorter because SA relocated to the city of ED2 for a month solely for the purpose of data collection. This focused approach enabled her to accumulate the necessary hours of observation in less than a month.

Furthermore, we intentionally avoided data collection during the second part of December to prevent the potential biases associated with holiday seasons, such as Christmas and national bank holidays. These periods can be atypical and could have skewed the data. Therefore, SA resumed and completed the observations in January, when regular ED activity was normalized.

It is important to note that the total hours of observation were our primary concern, rather than the length of the observation period per se. This approach ensured that we gathered a sufficient and comparable amount of data from each ED, allowing for a robust analysis despite the temporal differences.     

Pg 8, line 224-225 - I am not sure I understand how the data presented demonstrates that pharmacists were given the autonomy to expand the scope of their work?

Thank you for your comment. You are right, the data presented do not demonstrate that. Our intention was to convey that while the pharmacists had the opportunity to engage in a broad spectrum of activities, the reality was that they predominantly chose to focus on MedRec.
In the Methods section, we highlighted that the pharmacists were encouraged to explore and adapt to the new environment, finding their own ways to contribute to the team. This pragmatic  approach to the intervention is detailed in the last paragraph of section 2.2. of the manuscript. However, the findings indicate that despite this encouragement, the pharmacists’ activities were largely confined to MedRec.

We have further clarified this point in the discussion, exploring potential reasons for the pharmacists’ preference for MedRec over other activities. Additionally, I would like to specify that we had categories in place to register patient consultations, MedRew and another tasks. However, those categories were registered rarely.

Pg 8, line 226-232 - Suggest removing these lines as they are not currently supported by the data collected and don't add substantively to the stated aim of the project.

Thank you for your thoughtful suggestion regarding lines 226-232. We understand your concern. However, we believe that the inclusion of these lines is crucial for the contextualizing the pharmacists’ tasks within the scope of our study. The fact that the pharmacists primarily performed MedRec is indeed one of our key findings, and these sentences offer potential explanations for this outcome. We have carefully considered your recommendation and have concluded that retaining these lines adds value by providing insight into the pharmacists’ roles. Therefore, we kindly request the opportunity to keep this section in the manuscript.     

Pg 8, line 238-253 - Suggest moving this paragraph up to be second in the discussion section. 

Thank you for your suggestion, it has now been done. 

Pg 9, line 266-270 - Suggest removing these two sentences as they do not add to the stated aim of the paper.

Thank you for your suggestion to remove lines 266-270. We have considered your recommendation and understand the importance of ensuring that every part of our manuscript aligns with the stated aim of the paper.

The aim of our paper, as stated, is to investigate how pharmacists distributed their work during the intervention period in two of the EDs. We believe that the sentences in question are pertinent to this aim as they provide a discussion on the observed differences in tasks distribution between the two EDs. The sentences offer potential explanations for the variance observed, such as the pharmacists’ placement within the EDs, their ease of access to physicians and nurses, and the opportunities for consultation on complex cases. Additionally, the time spent by pharmacists in the ED compared to other wards is a significant factor that could influence their task distribution. We argue that examining the work tasks in both EDs involves exploring the differences between them. Therefor, we respectfully request to retain these lines as they contribute to an understanding of the pharmacists’ work task distribution and the factors influencing it. 

Reviewer 2 Report

Comments and Suggestions for Authors

Thank you for the interesting and very well written article on the distribution of pharmacists' work time in emergency departments.

I have just two minor comments:

Line 88: You talk about eleven pharmacists. However, results are only reported for ten. Was one pharmacist excluded? If I have not missed it, please add or correct the information.

Please check the manuscript again according to the journal's author guidelines: https://www.mdpi.com/journal/pharmacy/instructions

For example, the references in the text should be placed in square brackets. Please also adapt the citation style according to the guidelines.

Author Response

Dear reviewer,

Thank you for the time you dedicated to reviewing our manuscript. Your comments and kind words are deeply appreciated. 

Line 88: You talk about eleven pharmacists. However, results are only reported for ten. Was one pharmacist excluded? If I have not missed it, please add or correct the information.

Thank you for bringing this to our attention. We apologize for any confusion caused by the discrepancy in the number of pharmacists observed. As you correctly noted, one pharmacist was unable to participate in the observations due to parental leave. We have now clarified this in the manuscript by adding a sentence in 2.2. section of the Methods: “Among the eleven pharmacists integrated into the two EDs, we observed ten.” 

Please check the manuscript again according to the journal's author guidelines: https://www.mdpi.com/journal/pharmacy/instructions
For example, the references in the text should be placed in square brackets. Please also adapt the citation style according to the guidelines.

We appreciate your guidance on adhering to the journal's author guidelines. Following your suggestion, we have thoroughly reviewed the manuscript and adjusted the citation style to comply with the guidelines, ensuring that references are now placed in square brackets.

Additionally, we have conducted a comprehensive review of the entire manuscript to ensure all aspects conform to the journal's standards. If there are any further specific adjustments needed, we are more than willing to make the necessary changes to meet the requirements.

Reviewer 3 Report

Comments and Suggestions for Authors

Thank you for the opportunity to review your manuscript. Overall, i found your work to be very well written and very valuable in the ED space. Your introduction provided good context for the study and included a wide range of relevant references, which supported your overarching aim.  The methods were very well described and the choice of the WOMBAT approach was appropriate and well justified. This type of observational study can be difficult to achieve, but you have demonstrated a robust approach to data collection and analysis. The results were clearly presented and well described. I found the tables easy to read and they included information relevant to your aim. The discussion was also nicely aligned to your aim and provided a good, high level analysis of the results. You have clearly acknowledged the strengths and limitation of your study, including the potential inability to generalise these findings. However, I think that your work is highly valuable and adds to the current body of work around ED Pharmacists. Well done. One minor comment below.  

Line 231 - The word "Inclined" is probably not the best choice in this sentence. I would suggest rewording.  

Author Response

Dear reviewer,

I sincerely appreciate your kind words and valuable assessment of our manuscript. Your feedback is highly valued and we are grateful for your thoughtful review. 

Line 231 - The word "Inclined" is probably not the best choice in this sentence. I would suggest rewording. 

Thank you for your suggestion. The word has now been replaced: "This is further underscored by the fact that MedRec is often perceived by ED physicians as both time-consuming and a detective job (5), which may have motivated physicians to readily delegate this responsibility to the pharmacists."